# Design and implementation of digital literacy training programme: Findings of a quasi-experimental study from rural India

Aparajita Gogoi[1], Mercy Manoranjini[1]*, Mamta Gupta[2]

1 Centre for Catalyzing Change, New Delhi, India, 2 Unit of Epidemiology & Biostatistics, Alchemist Research & Data Analysis, Chandigarh, India

* mmanoranjini@c3india.org

## Abstract

In India, digital divide is evident among rural sections of the society among women. For bridging this gap, the physical access to information and communications technology (ICT) along with training to harness these skills is important. This would empower the girls to gain more control over their health, finances, and safety, and can more freely assert their voice and agency. The project aimed to measure the knowledge and skills about the digital tools and its impact in different spheres of life (status of schooling, uptake of government schemes and mobility) for the adolescent girls before and after the implementation of the digital literacy training programme (DLTP) in rural India. The project was implemented in eight blocks of District Gumla of Jharkhand for three years from 2017-2020. The evaluations were conducted in three terms that is, baseline, midline and endline by using quasi-experimental research design. A quantitative questionnaire was developed to collect data from school dropout girls of 10–19 years. The sample coverage for intervention and comparison arm was 314 and 272 at the baseline, 318 and 260 at the midline, and 402 vs 202 in the endline. A state-of-the-art training curriculum was designed covering various components of digital literacy (hardware, software, applications, internet, emailing, social media, cyber security, education and career opportunities in the digital space). A team of facilitators were provided a laptop and a pico projector to conduct trainings in the intervention blocks. Majority of the participants were unmarried, lived along their parents and had ever attended school in both intervention and comparison arm. There was no significant difference in the proportion of girls having digital literacy score above median between intervention and comparison arm at the baseline. At the midline, the proportion of these girls with above median score was significantly higher in the intervention arm [n=203, 63.8%] as compared to the comparison arm [n=107, 41.2%]. Similarly, at the endline, the proportion of girls with higher median score for digital literacy increased significantly in the intervention arm [n=362, 90%] as compared to comparison arm [n=84, 41.6%]. The proportion of girls continuing education, physical access to ICT devices at home, exposure to mass media and perceived physical security increased in the intervention arm at the midline and endline as compared to comparison arm in the baseline. A significantly higher proportion of girls had knowledge about government schemes in the intervention arm as compared to comparison arm at the endline (p<0.05). We conclude

**Data availability statement:** All the relevant data are within the paper.

**Funding:** The author(s) received no specific funding for this work.

**Competing interests:** The authors have declared that no competing interests exist.

that such tailor-made training programs which are weaved within cultural contexts can effectively bridge the digital gaps in resource scare settings. Along with this, they have the potential to bridge the literacy and economic gaps also within the community.

---

## Author summary

In rural India, a digital divide exists, particularly among women, hindering their access to information and opportunities. To address this, a project was implemented in Jharkhand from 2017 to 2020, focusing on digital literacy training for adolescent girls. The aim was to assess the impact of this training on their knowledge and skills related to digital tools, and its influence on areas like schooling, government schemes, and mobility. Using a quasi-experimental design, the study evaluated 314 girls in the intervention arm and 272 in the comparison arm. The results showed significant improvements in the digital literacy scores of girls in the intervention group, with a higher proportion gaining skills by midline and endline evaluations. Additionally, girls in the intervention group reported better access to ICT devices, improved education continuation, increased knowledge of government schemes, and greater perceived safety. The study concluded that culturally tailored digital literacy programs can bridge digital, literacy, and economic gaps.

## Introduction

Digital technology has become integral to our daily lives, impacting how we access information, communicate, and innovate. UNESCO defines digital literacy (DL) as the ability to use digital technologies to access, manage, create, and evaluate information for purposes such as employment and entrepreneurship [1]. It has empowered children and adolescents by providing access to online education, health information, job opportunities, and financial services especially during and after COVID-19 pandemics [2–4]. However, unequal access to technology and skills can create a digital divide, leaving some individuals at a disadvantage [2,5,6].

Globally, around 3.7 billion people remain unconnected, with the majority being women, girls, and people in remote, rural areas with low education and income [7]. A 2018 National Sample Survey found that only 4% of rural households owned a computer, and just 7% of women could operate one [8]. The National Family Health Survey-5 revealed that 46.6% of women owned a mobile phone, compared to 72.3% of men, and only 24.6% of rural women had used the internet, compared to 48.7% of men [9]. Smartphone usage in India rose from 23% in 2016 to 54% in 2021, but the gendered digital divide remains at 40.4% [10]. A study conducted during the COVID-19 lockdown in four Indian states showed that adolescent boys had more access to newspapers, radios, and mobile phones, with higher usage of social media and online learning platforms compared to girls [11]. This concludes that many people, especially women and rural populations, still lack access to basic digital tools and the skills to use them.

A recent study of 11,707 young unemployed women trained in digital literacy under NDLM or DISHA found that basic ICT training improved their skills, leading to better educational and employment opportunities [12]. The Government of India launched the "Digital India Mission" on 1st July 2015, aimed to create a digitally empowered society and knowledge economy by bridging the digital divide. Similarly, the Ministry of Electronics & Information Technology (MeitY) has implemented initiatives like the National Digital Literacy Mission (NDLM), Digital Saksharta Abhiyan (DISHA), and Pradhan Mantri Gramin Digital Saksharta

Abhiyan (PMGDISHA) to enhance digital literacy [13] In line with this, NITI Aayog introduced the "Strategy for New India @75" to bridge the digital divide [14]. The Digital India Report (2020) highlighted that though infrastructure has improved with these initiatives, the challenge of reaching remote, tribal, and rural regions persists, hindering the true reach of digital literacy programs. Also, high dropout rates and inconsistent training quality due to lack of proper follow-ups remains a challenge in implementing these programs.

With an aim to address the existing gaps, Centre for Catalyzing Change (C3) designed and implemented its "Akanksha project: the desire to learn" in 2017 among tribal and disadvantaged caste communities [15]. One of the objectives of this project was to nurture a cadre of empowered adolescent girls who would have access to digital tools, be digitally literate, confident in accessing opportunities via digital platforms, and equipped to make better life choices. This study demonstrates the development and implementation of a Digital Literacy Training Program (DLTP) to measure and compare the knowledge and skills about the digital tools, its importance and practical usage among adolescent girls during three phases of implementation.

## Methodology

This study was conducted using a quasi-experimental research design among adolescent girls aged 14–19 years, across eight community development blocks of District Gumla in Jharkhand, India over a period of three years from 2017-2020 [16]. District Gumla is situated in the southwest of Jharkhand, is predominantly inhabited by tribal and marginalized caste communities. The region exhibits a significant gender disparity in literacy rates, with young girls facing considerable obstacles in completing their education. This is due to a combination of local unrest and deeply entrenched patriarchal barriers, which restrict safe spaces for young girls to formally organize and engage in learning activities. Consequently, Gumla was identified as a critical site for initiating the Akanksha project, with a targeted focus on promoting digital literacy among girls in the region.

At the outset, a baseline evaluation was conducted among adolescent girls to assess the need for digital literacy from their perspectives in 2017. The evaluation focused on determining the optimal timing and location for training delivery, identifying the necessary infrastructure, selecting the appropriate mode of content delivery, understanding participants' expectations from the program, and exploring their plans for utilizing the acquired skills in the future.

The Digital Literacy Training Program (DLTP) was developed in consultation with community leaders, school teachers, and village volunteers. The program addressed three key challenges: a) digital access, b) digital literacy, and c) life skills.

a. **Digital access:** To tackle the challenge of digital access, sensitization sessions were conducted for family members of the girls to encourage them to support digital literacy initiatives. During the baseline discussions, the preferred times and locations for the training sessions were explored with both girls and their families. It was found that the participants were most comfortable attending training sessions at Anganwadi centers either in the morning before school or in the afternoon after school. Additionally, community interactions revealed ongoing efforts by the central and state governments, as well as local NGOs, to promote digital education for adolescents. The program sought to integrate and leverage these existing resources, such as computers and internet connections, for exclusive training of girls.

b. **Digital literacy:** To empower adolescents with digital skills, a state-of-the-art training curriculum was designed in collaboration with the Digital Empowerment Foundation (DEF). The curriculum combined theoretical knowledge, interactive videos, and activities. A comprehensive manual covering various aspects of digital media—such as hardware, software, applications, the internet, email, social media, cybersecurity, and digital education and career opportunities—was provided to facilitators. The curriculum also included digital

life skills education, with a focus on developing negotiation skills. Topics like cyberbullying and internet safety were integral to the program. Capacity-building workshops and training for facilitators were organized, and each facilitator was equipped with a laptop and pico projector to deliver the training.

c. **Life skills:** In addition to digital literacy, the program focused on imparting essential life skills to the girls, including critical thinking, negotiation, reproductive and sexual health knowledge, agency-building, and financial literacy.

As it was a quasi-experimental study, it was envisioned to develop and implement a Digital Literacy Training Program in the intervention arm and measure its effect over a period of three years with the comparator arm at the baseline (in 2017), midline (2019) and endline (2020). We provided training and support around life skills education, sexual and reproductive health (SRH) awareness and gender equity to both intervention and comparator blocks as a part of our social responsibility.

## Sample size

The sample coverage for intervention and comparison arm was 314 and 272 at the baseline, 318 and 260 at the midline, and 402 vs 202 in the endline. For the intervention arm, the sample size of 300 was computed using a standard formula to estimate proportions for measuring the present knowledge level at a 95% confidence level and 5% margin of error. A sample size of 200 for the comparison arm was calculated based on a formula for measuring change. The key parameters considered were: $p_1$ = 12% (correctly answer at least half the questions on digital literacy), $p_2$ = expected level- assuming a 10% change upwards, $Z_1$-$\alpha$ = Constant set according to the confidence level, for 95%, this value is 1.96 and $Z_1$-$\beta$ = constant set according to the power of a study, for 80%, this value is 0.84.

## Sampling technique

A multistage sampling was used. In the first stage, out of 12 blocks of District Gumla, 8 were selected (six intervention and two comparator) as these represented the field area of C3 in Jharkhand. Of the eight community development blocks in District Gumla, six were purposively selected as intervention blocks (Gumla Sadar, Sisai, Kamdara, Basia, Palkot and Chainpur) and rest two were the comparison (non-intervention) blocks that is, Ghagra and Raidih on the suggestions of district administration.

In the second stage, the villages were selected by proportional random sampling based on population of the block and village. For this a list of villages was taken from Block Development Officer under every block. Finally, all the adolescent girls from each selected village were listed alphabetically and were selected using systematic random sampling. An average of 15 adolescent girls were interviewed from every village.

## Data collection tool

A semi-structured questionnaire was developed in Computer-Assisted Personal Interviews (CAPI) to collect data in- person by interviewing participants at baseline, midline and endline; both in the intervention arm and comparison arm. Then, the face and content validity of these questionnaires was established by the experts for collecting data about the sociodemographic characteristics, their household possessions, status and opinions about existing Digital Literacy Training Programmes (DLTPs) in vicinity, their level of exposure to media, existing knowledge and practice about basic computer functions, internet, digital safety and cyberbullying, and awareness regarding various government schemes.

## Data collection method

The houses of the girls were approached with the support of community health workers, village headman and teachers. Data was collected after obtaining assent or consent of the parents. COVID-19 protocols were strictly followed during the pandemic period. While the study aimed to interview the same adolescent girls during the baseline and end line evaluations, this was challenging due existing socio-cultural factors in the region. In both, the intervention and comparison groups, many of the girls interviewed during the baseline study had either migrated or got married and moved away. Others were unavailable because they were traveling or had long work or study hours. In such cases respondents with similar age and socio-economic profiles from the same or the adjacent village were used as replacements in the midline and endline.

## Data analysis

The information gathered on independent variables through the questionnaire was used to calculate various scores following standardized methodology:

i. **The household possession score** was calculated using the District Household Survey-7 methodology as a proxy to economic status [17]. This score consisted of the total of four sub-scores that is, household effect, availability of means of transport, agricultural land and livestock.

ii. **Financial independence score:** This variable was computed to assess whether the girl was able to utilize her earnings by herself or not.

iii. **Perceived physical security score:** This referred to how a girl perceived about her safety in going out to market, to community place etc as a proxy indicator to her mobility.

iv. **Physical access to IT devices and internet score:** This composite variable was computed to assess the accessibility of a girl to smart phone, computer or laptop and internet. These are the essential inputs of digital ecosystem.

v. **Intensity of exposure to mass media score:** This composite score referred to frequency (daily, occasionally, and never) at which the mass media platforms like television, newspaper and radio are accessible to the participant.

vi. **Awareness and uptake of government schemes score:** This composite variable evaluated the existing knowledge and uptake of various government schemes like voter ID, AADHAR card etc among young girls. This can be considered as a proxy marker for the impact of DLTP.

The outcome of interest in this study was digital literacy which was operationally defined as "ability to use information and communication technologies to find, evaluate, create, and communicate using both cognitive and technical skills". In the analysis, the questions assessing DL among the participants were combined according to elements of digital literacy framework for calculating DL scores. The five basic elements of this framework were evaluated as described below [18]:

i. **Technical:** The questions capturing information on the foundational digital skills, which includes powering on and off devices, accessing tools and applications from devices, using the mouse and touchpad, and troubleshooting were grouped under this component.

ii. **Civic:** This element is similar to traditional citizenship where individuals have rights and responsibilities that need to be respected. Online safety, building identity, managing reputation, etiquette, and participation are a part of the civic element.

iii. **Communicative:** In this, the individuals share a variety of resources and materials with others using various platforms like writing a professional email or memorandum, posting images, and adding comments on social media.

iv. **Computational Thinking:** It refers to knowledge and practice of leveraging digital media and technology to solve a problem and use critical thinking skills. Like creating a spreadsheet for budgeting, analysing a poem for structure, and putting together an agenda for a work meeting.

v. **Information literacy:** It highlights an individual's ability to search, identify, and validate the information like using digital platforms for shopping and price comparison.

The data were entered and analysed using SPSS 23.0 version (IBM Corp. Released 2015. IBM SPSS Statistics for Windows, Version 23.0. Armonk, NY: IBM Corp.). Univariate analysis was used to describe the independent variables (sociodemographic and socioeconomic characteristics) as their numbers and percentages. For other independent variables comprising various scores, a composite score was calculated for every respondent as per the methodology presented above. The outcome variable constituted of individual components and composite score for DL was presented as median with interquartile range (IQR). Numbers and percentages of girls in two categories of DL (one below and other above the median) were calculated to present the findings for composite variables. Chi-square test of significance (P < 0.05) was used to compare the independent variables and DL scores among intervention and comparison arm at baseline, midline and endline.

## Ethical considerations

The protocol and research tools were reviewed by the Ethical Review Board of Sigma-IRB and approved by the Institutional Review Board of India in accordance with the compliance of the Title 45, Code of Federal Regulations; Sub-Part A (Common Rule) of NIH dated December 5, 2018, with IRB number 10045/IRB/D/18–19. The data collection tools were administered only after obtaining the participant's written consent after explaining the purpose of the project. The participation was voluntary. The confidentiality and privacy of the participants has been maintained.

## Results

### Socio-demographic and socio-economic characteristics

The number of girls aged 14–19 years covered in the intervention and comparison arm corresponded to 314 and 272 at the baseline, 318 and 260 at the midline, and 402 vs 202 in the endline, respectively (Table 1). More than 90% of girls were unmarried at the evaluations in both arms. The education status of fathers corresponded to no formal schooling among 41% and 38% in the baseline, 33% and 42% in the midline; and 31%, 41% in the endline between the intervention and comparison arm, respectively. Mothers of most adolescent girls had no formal education in both intervention and comparison arms (73% and 69% at the baseline, 70% each at the midline, and 46% and 60% at the endline phase).

At baseline, around 54% and 67% of adolescent girls reported having household or work responsibility in the intervention and comparison arm, respectively. This proportion reduced to 30% and 38% at the midline, 31% and 23% at the endline in the intervention and comparison arms, respectively. Around 53.2% in the intervention arm and 50% in the comparison arm had household possession scores of ≥ 4 at the baseline. This proportion increased to

**Table 1. Socio-demographic and socio-economic characteristics of participants.**

| Variable | Attributes | Intervention BL n=314 N (%) | Comparator BL n=272 N (%) | P value | Intervention ML n=318 N (%) | Comparator ML n=260 N (%) | P value | Intervention EL n=402 N (%) | Comparator EL n=202 N (%) | P value |
|---|---|---|---|---|---|---|---|---|---|---|
| School ever attended by girls | Yes | 312 (99.4) | 272 (100) | 0.19 | 314 (98.7) | 256 (98.5) | 0.77 | 400 (99.5) | 202 (100) | 0.31 |
| | No | 2 (0.6) | 0 (0) | | 4 (1.3) | 4 (1.5) | | 2 (0.5) | 0 (0) | |
| Marital status | Never married | 305 (97.1) | 251 (92.3) | **0.01** | 309 (97.2) | 247 (95) | 0.17 | 401 (99.8) | 202 (100) | 0.48 |
| | Married | 9 (2.9) | 21 (7.7) | | 9 (2.8) | 13 (5) | | 1 (0.2) | 0 (0) | |
| Father's educational status | No formal schooling | 122 (41.5) | 95 (38.6) | 0.06 | 106 (33.3) | 111 (42.7) | **0.01** | 125 (31.1) | 81 (40.1) | **0.02** |
| | Till metric | 138 (46.9) | 135 (54.9) | | 170 (53.5) | 133 (51.2) | | 203 (50.5) | 99 (49) | |
| | Above metric | 34 (11.6) | 16 (6.5) | | 42 (13.2) | 16 (6.2) | | 74 (18.4) | 22 (10.9) | |
| Father's occupation | Unemployed | 20 (6.8) | 14 (5.7) | **0.02** | 11 (3.5) | 12 (4.6) | **0.01** | 104 (25.9) | 28 (13.9) | **0.00** |
| | Farming on own land | 216 (73.5) | 150 (61) | | 209 (65.7) | 133 (51.2) | | 119 (29.6) | 95 (47) | |
| | Unskilled Labour | 27 (9.2) | 36 (14.6) | | 36 (11.3) | 55 (21.2) | | 16 (4) | 14 (6.9) | |
| | Business | 18 (6.1) | 29 (11.8) | | 26 (8.2) | 25 (9.6) | | 90 (22.4) | 26 (12.9) | |
| | Government/Private services | 12 (4.1) | 14 (5.7) | | 12 (3.8) | 14 (5.4) | | 37 (9.2) | 13 (6.4) | |
| | Father not alive | 1 (0.3) | 3 (1.2) | | 24 (7.5) | 21 (8.1) | | 36 (9) | 26 (12.9) | |
| Mother's educational status | No formal schooling | 228 (73.5) | 184 (69.2) | 0.39 | 221 (69.5) | 180 (69.2) | 0.69 | 185 (46) | 122 (60.4) | **0.00** |
| | Upto metric | 77 (24.8) | 79 (29.7) | | 89 (28) | 76 (29.2) | | 192 (47.8) | 77 (38.1) | |
| | Above metric | 5 (1.6) | 3 (1.1) | | 8 (2.5) | 4 (1.5) | | 25 (6.2) | 3 (1.5) | |
| Mother's occupation | Unskilled labour | 10 (3.2) | 10 (3.8) | **0.03** | 169 (53.1) | 172 (66.2) | **0.00** | 122 (30.3) | 42 (20.8) | **0.00** |
| | Farming on own land | 121 (44.0) | 77 (29) | | 105 (33) | 54 (20.8) | | 197 (48.7) | 115 (56.9) | |
| | Shop owner/business | 7 (2.3) | 10 (3.8) | | 3 (0.9) | 5 (1.9) | | 53 (13.2) | 14 (6.9) | |
| | Government/Private service | 13 (4.2) | 25 (9.4) | | 17 (5.3) | 9 (3.5) | | 25 (6.2) | 1 (0.5) | |
| | Mother not alive | 159 (51.3) | 144 (54.1) | | 24 (7.5) | 20 (7.7) | | 5 (1.2) | 9 (4.5) | |
| Support family financially | Yes | 143 (45.5) | 89 (32.7) | **0.01** | 226 (71.1) | 162 (62.3) | **0.03** | 277 (68.9) | 159 (78.7) | **0.01** |
| | No | 171 (54.5) | 183 (67.3) | | 92 (28.9) | 98 (37.7) | | 125 (31.1) | 43 (21.3) | |
| Household possessions related to ICT like mobile, computer, internet availability etc | ≤ 2 | 273 (86.9) | 213 (78.3) | **0.01** | 272 (85.5) | 236 (90.8) | 0.05 | 219 (54.5) | 162 (80.2) | **0.00** |
| | >2 | 41 (13.1) | 59 (21.7) | | 46 (14.5) | 24 (9.2) | | 183 (45.5) | 40 (19.8) | |
| Means of transport | ≤ 1 | 247 (78.7) | 198 (72.8) | 0.09 | 239 (75.2) | 194 (74.6) | 0.88 | 220 (54.7) | 151 (74.8) | **0.00** |
| | > 1 | 67 (21.3) | 74 (27.2) | | 79 (24.8) | 66 (25.4) | | 182 (45.3) | 51 (25.2) | |
| Total_household_possessions | ≤ 4 | 147 (46.8) | 135 (49.6) | 0.49 | 185 (58.2) | 188 (72.3) | **0.00** | 133 (33.1) | 111 (55) | **0.00** |
| | > 4 | 167 (53.2) | 137 (50.4) | | 133 (41.8) | 72 (27.7) | | 269 (66.9) | 91 (45) | |
| Household/work responsibility | Yes | 171 (54.5) | 183 (67.3) | **0.01** | 94 (29.6) | 99 (38.1) | **0.02** | 125 (31.1) | 47 (23.3) | **0.03** |
| | No | 143 (45.5) | 89 (32.7) | | 224 (70.4) | 161 (61.9) | | 277 (68.9) | 155 (76.7) | |

66.9% and 45% at the endline in the intervention and comparison arm, respectively. The proportion of households having a score of >1 for physical access to IT devices increased from 33.8% to 83% in the intervention arm and 35% to 64% in the comparison arm from baseline to endline.

## Comparison of digital literacy- intervention arm vs comparison arm

Table 2 represents the median scores for the elements of DL and Table 3 compares the proportion the adolescent girls having digital literacy on its various elements between intervention and control arm at three evaluations.

1) At the baseline, the median scores and interquartile range for the knowledge on the technical element of DL were 1 (0–2) and 0 (0–2), respectively in the intervention arm and comparison arm. In the intervention arm, the score increased to 2 (0–5) and 9 (5–13) in the midline and endline phases, respectively. However, it remained constant at 0 in the comparison arm in both phases. This indicates the element of DL, that is, knowledge on the technical element increased from baseline to midline and endline in the intervention arm (Table 2). The proportion of adolescent girls in the intervention arm with a score of >1 increased from 41% at baseline to 57% at the midline; that further increased to 94% in the endline. But the proportion of adolescent girls with a score of more than 1 remained 23–25% in the comparison arm throughout. And this increase in the intervention arm as compared to comparison arm was found to be significant ($p<0.05$) in all three evaluations (Table 3).

2) Similarly, the median score for the civic element of DL increased in the intervention arm from 0 (0–1) at baseline to 2(1–3) and 3 (2–4) in the midline and endline phases respectively. Whereas, in the comparison arm it was 0 (0–1) in the baseline and remained constant at 1 in the midline and endline phases (Table 2). At the baseline- only 18(5.7%) and 15 (5.5%) adolescent girls obtained a score of >1 in both intervention and comparison arms, respectively. The proportion increased to 58% and 46% in both arms at the midline. And at the endline, it increased to 79% and 46% in the intervention and comparison arm, respectively. This increase in the knowledge aspect of civic DL between intervention arm and comparison arm was found to be significant at the midline and endline ($p<0.05$) (Table 3).

3) The median scores for the knowledge aspect of the communicative element of DL in both intervention and comparison arms were the same at 0 in the baseline and midline. It increased to 1 in both arms, at the endline (Table 2). The proportion of adolescent girls having knowledge about writing emails, or use social media platforms like youtube was significantly higher in the intervention arm as compared to control arm at the midline and endline that is after roll out of DLTP as reflected in Table 3.

**Table 2. Median scores for elements of digital literacy, and its impact on the knowledge and awareness of government schemes among adolescent girls.**

| Composite variables | Baseline | | Midline | | Endline | |
|---|---|---|---|---|---|---|
| | Intervention (n=314) | Comparator (n=272) | Intervention (n=318) | Comparator (n=260) | Intervention (n=402) | Comparator (n=202) |
| | Median (IQR) | Median (IQR) | Median (IQR) | Median (IQR) | Median (IQR) | Median (IQR) |
| Knowledge aspect of technical DL | 1 (0 – 2) | 0 (0–2) | 2 (0–5) | 0 (0–1) | 9 (5–13) | 0 (0–1) |
| knowledge aspect of Civic DL | 0 (0–1) | 0 (0–1) | 2 (1–3) | 1 (0–3) | 3 (2–4) | 1 (1–3) |
| Knowledge aspect of communicative DL | 0 (0–0) | 0 (0–0) | 0 (0–1) | 0 (0–1) | 1 (1–1) | 1 (0–1) |
| Practice aspect of communicative DL | 1 (1–2) | 1 (1–2) | 2 (1–2) | 2 (1–2) | 3 (2–4) | 2 (1–4) |
| Knowledge aspect of computational thinking DL | 1 (0–1) | 1 (0–1) | 0 (0–1) | 0 (0–1) | 0 (0–1) | 0 (0–1) |
| Knowledge aspect of investigative DL | 0 (0–0) | 0 (0–0) | 0 (0–1) | 0 (0–0) | 1 (0–1) | 0 (0–0) |
| Total score DL | 10 (8–13) | 10 (8–12) | 16 (12–21) | 12 (10–16) | 27 (20–33) | 12 (9–17) |
| Knowledge and uptake of government schemes | 5 (4–6) | 4 (3–6) | 7 (5–7) | 7 (5–7) | 7 (5–7) | 5 (4–7) |

**Table 3.** Comparing elements of digital literacy and awareness of government schemes among adolescent girls between intervention and comparison blocks.

| Composite variables | Scores | Baseline | | | Midline | | | Endline | | |
|---|---|---|---|---|---|---|---|---|---|---|
| | | Intervention | Comparator | | Intervention | Comparator | | Intervention | Comparator | |
| | | n= 314 | n= 272 | p value | n= 318 | n= 260 | p value | n= 402 | n= 202 | p value |
| Knowledge aspect of technical DL | >1 | 131 (41.7) | 70 (25.7) | **0.00** | 184 (57.9) | 63 (24.2) | **0.00** | 378 (94) | 48 (23.8) | **0.00** |
| | ≤ 1 | 183 (58.3) | 202 (74.3) | | 134 (42.1) | 197 (75.8) | | 24 (6) | 154 (76.2) | |
| knowledge aspect of civic DL | > 1 | 18 (5.7) | 15 (5.5) | 0.9 | 185 (58.2) | 119 (45.8) | **0.00** | 316 (78.6) | 94 (46.5) | **0.00** |
| | 0 or 1 | 296 (94.3) | 257 (94.5) | | 133 (41.8) | 141 (54.2) | | 86 (21.4) | 108 (53.5) | |
| Knowledge aspect of communicative DL | > 0 | 22 (7) | 16 (5.9) | 0.58 | 136 (42.8) | 65 (25) | **0.00** | 341 (84.8) | 101 (50) | **0.00** |
| | ≤ 0 | 292 (93) | 256 (94.1) | | 182 (57.2) | 195 (75) | | 61 (15.2) | 101 (50) | |
| Practice aspect of communicative DL | >2 | 47 (15) | 35 (12.9) | 0.46 | 61 (19.2) | 22 (8.5) | **0.00** | 287 (71.4) | 87 (43.1) | **0.00** |
| | ≤ 2 | 267 (85) | 237 (87.1) | | 257 (80.8) | 238 (91.5) | | 115 (28.6) | 115 (56.9) | |
| Knowledge aspect of computational thinking DL | 1 | 181 (57.6) | 163 (59.9) | 0.57 | 126 (39.6) | 81 (31.2) | **0.03** | 135 (33.6) | 75 (37.1) | 0.38 |
| | 0 | 133 (42.4) | 109 (40.1) | | 192 (60.4) | 179 (68.8) | | 267 (66.4) | 127 (62.9) | |
| Knowledge aspect of investigative DL | 1 | 45 (14.3) | 52 (19.1) | 0.12 | 96 (30.2) | 24 (9.2) | **0.00** | 266 (66.2) | 25 (12.4) | **0.00** |
| | 0 | 269 (85.7) | 220 (80.9) | | 222 (69.8) | 236 (90.8) | | 136 (33.8) | 177 (87.6) | |
| Total_score_DL | >13 | 72 (22.9) | 47 (17.3) | 0.1 | 203 (63.8) | 107 (41.2) | **0.00** | 362 (90) | 84 (41.6) | **0.00** |
| | ≤ 13 | 242 (77.1) | 225 (82.7) | | 115 (36.2) | 153 (58.8) | | 40 (10) | 118 (58.4) | |
| Knowledge and uptake of government schemes | > 6 | 63 (20.1) | 39 (14.3) | 0.07 | 209 (65.7) | 170 (65.4) | 0.93 | 242 (60.2) | 66 (32.7) | **0.00** |
| | ≤ 6 | 251 (79.9) | 233 (85.7) | | 109 (34.3) | 90 (34.6) | | 160 (39.8) | 136 (67.3) | |

4) The median scores for the practical aspect of the communicative element of DL in both intervention and comparison arms were the same in the baseline and midline. It increased to 3 and 2 in both arms, respectively at the endline (Table 2). The proportion of adolescent girls possessing the ability to write emails, or use social media platforms like youtube was significantly higher in the intervention arm as compared to control arm at the midline and endline that is after roll out of DLTP as reflected in Table 3.

5) In both the intervention and comparison arm there was a decrease in the median score in of knowledge aspect of computational thinking element of DL from 1(0–1) at baseline to 0(0–1) in the midline and endline (Tables 2 and 3)

6) In the comparison arm the median score was 0 for the knowledge aspect of investigative element of DL in all three phases. It was also 0 in the intervention arm at baseline and midline, and later increased to 1(0–1) in the endline phase. Only 14% and 19% of the adolescent girls obtained a score of 1 at the baseline in both the arms, respectively(p=0.12). However, for the intervention arm, the proportion increased to 30% and 66% at the midline and endline with significant difference from the comparison arm (p=0.000). (Tables 2 and 3).

7) The total DL score was same in both the arms at the baseline. It increased to 16 (12–21) and 12 (10–16) in the intervention arm and comparison arm, respectively at the midline. And further increased to 27 (20–33) in the intervention arm but remained constant in the comparison arm at the endline (Table 2). The proportion of adolescent girls having overall digital literacy more than the median score were same in both the arms at the baseline. At the midline, the proportion of these girls increased to 64% in the intervention arm and 41% in the comparison arm and this difference in the proportions of two arms was found to be statistically significant. Similarly, a significant difference was observed in the proportion of girls having high digital literacy (with a median score more than 13) in the intervention arm (90%) as compared to comparison arm (41%) at the endline.

The distribution of perceived physical security and exposure to mass media scores of adolescent girls in the intervention and comparison arms was not significantly different in baseline and midline. Around 29% of girls scored >10 in the intervention arm as opposed to 9% in the comparison arm for the perceived physical security parameter. And this difference was found to be statistically significant between the two arms in the endline (p=0.00).

Median scores for knowledge and uptake of government schemes in the intervention arm were 5(4–6) at the baseline. It remained the same at 7(5–7) in the midline and endline phase. In the comparison arm the median score was 4 (3–6) at the baseline, which increased to 7(5–7) in the midline; however, it decreased to 5(4–7) at the endline. The proportion of adolescent girls having knowledge of government schemes and who utilized them (score of >6) was 20% in the intervention arm and 14% in the comparison arm at the baseline (Table 4).

## Discussion

The present study has attempted to measure the knowledge and skills about the digital literacy and its components before and after the implementation of a digital literacy training program among the adolescent girls of Gumla District in Jharkhand of India using a quasi- experimental design. Gumla is one of the 24 districts in the state of Jharkhand having undulating terrain with a forest cover of more than 28%. Majority of population (>68%) is tribal and resides in rural areas. These are categorized as Particularly Vulnerable Tribal Groups (PVTGs) due to lack of social and economic progress. Given the geographical, socio-economic background and only a few non-governmental organizations (NGOs) working in Gumla, it is one of the aspirational districts of Jharkhand.

Therefore, it becomes a social responsibility to mainstream and link these populations with the outer world through digital literacy so that they are able to utilize the government schemes effectively. As a first step, along with socio-demographic-economic characteristics, the access to digital devices, perceived social security, awareness about the government schemes and

**Table 4. Changes in basic personal and household characteristics from baseline to endline.**

| Variable | Attributes | Intervention BL n=314 N (%) | Comparator BL n=272 N (%) | P value | Intervention ML n=318 N (%) | Comparator ML n=260 N (%) | P value | Intervention EL n=402 N (%) | Comparator EL n=202 N (%) | P value |
|---|---|---|---|---|---|---|---|---|---|---|
| Does your household have any of the Cards? | No | 53 (16.9) | 63 (23.2) | **0.03** | 35 (11) | 47 (18.1) | 0.05 | 68 (16.9) | 13 (6.4) | **0.00** |
| | BPL/Antyodya | 174 (55.4) | 156 (57.4) | | 236 (74.2) | 179 (68.8) | | 329 (81.8) | 189 (93.6) | |
| | APL | 87 (27.7) | 53 (19.5) | | 47 (14.8) | 34 (13.1) | | 5 (1.2) | 0 (0) | |
| Education continued | Yes | 230 (73.7) | 181 (66.5) | 0.06 | 245 (78) | 169 (66) | **0.00** | 384 (96) | 174 (86.1) | **0.00** |
| | No | 82 (26.3) | 91 (33.5) | | 69 (22) | 87 (34) | | 16 (4) | 28 (13.9) | |
| Financial Independence | No | 194 (61.8) | 180 (66.2) | 0.27 | 196 (61.6) | 191 (73.5) | **0.00** | 256 (63.7) | 88 (43.6) | **0.00** |
| | Yes | 120 (38.2) | 92 (33.8) | | 122 (38.4) | 69 (26.5) | | 146 (36.3) | 114 (56.4) | |
| Perceived physical security | ≤ 10 | 120 (38.2) | 111 (40.8) | 0.52 | 146 (45.9) | 106 (40.8) | 0.21 | 286 (71.1) | 182 (90.1) | **0.00** |
| | > 10 | 194 (61.8) | 161 (59.2) | | 172 (54.1) | 154 (59.2) | | 116 (28.9) | 20 (9.9) | |
| Intensity of exposure to mass media | ≤ 2 | 217 (69.1) | 169 (62.1) | 0.08 | 262 (82.4) | 222 (85.4) | 0.33 | 262 (65.2) | 165 (81.7) | **0.00** |
| | >2 | 97 (30.9) | 103 (37.9) | | 56 (17.6) | 38 (14.6) | | 140 (34.8) | 37 (18.3) | |
| Physical access to IT devices | ≤ 1 | 208 (66.2) | 177 (65.1) | 0.77 | 232 (73) | 211 (81.2) | **0.02** | 68 (16.9) | 73 (36.1) | **0.00** |
| | > 1 | 106 (33.8) | 95 (34.9) | | 86 (27) | 49 (18.8) | | 334 (83.1) | 129 (63.9) | |
| Knowledge and uptake of government schemes | > 6 | 63 (20.1) | 39 (14.3) | 0.07 | 209 (65.7) | 170 (65.4) | 0.93 | 242 (60.2) | 66 (32.7) | **0.00** |
| | ≤ 6 | 251 (79.9) | 233 (85.7) | | 109 (34.3) | 90 (34.6) | | 160 (39.8) | 136 (67.3) | |

digital literacy among were evaluated in both the intervention and comparison arms. Then the resources were mobilized to meet the access needs of the girls, and a training curriculum was designed to enable them in using digital platforms. The long-term goal behind rolling out this DLTP was to empower the girls to gain more control over their health, finances, and safety, and can more freely assert their voice and their agency.

The extant literature search indicated that most of the studies on DL pertain to conceptual critiques and reports on projects that have attempted to nurture these abilities and attributes [5]. These studies present the outcomes of DLTP in terms of improving knowledge about health, education, employment, marriage, and pregnancy before and after the intervention [5]. However, only a few studies have been conducted to assess the impact of community- level interventions on levels of DL before and after DLTP [2,19–22].

The review by Choudhary and Bansal (2022) clearly demonstrated that 60% of the studies targeting to improve DL have been conducted in developed countries. In most of these studies, the focus was on improving the technical elements of DL followed by information DL and communicative DL. Only five studies out of 86 studies in their review focussed on safety element of DL, three studies on problem solving element and only two studies for developing career competencies. In this study, we incorporated all five elements of digital literacy (DL) into the training curriculum, which constitutes both the strength and novelty of our approach. We systematically measured and documented changes across each of these five elements, as outlined in the DL framework in the methodology section [5].

According to a study conducted on 11,707 females in India, the digital literacy impacts not only the opportunity-seeking, be it educational or employment but also acts as a catalyst for perceived competence [12]. Similar findings have been reported many studies [22–25]. In a study among tribal area of Kerala, an educational model was used to train 1000 adults using an integrated curriculum via tablet-based applications. It showed that such an intervention may help in overcoming existing challenges, thereby improving digital as well as life skills.

In our study, we found that there was a statistically significant increase in DL and it's individual elements from baseline to midline and then to endline in the intervention arms as compared to comparison blocks (Tables 2 and 3). And correspondingly, a significant increase in the proportion of adolescent girls continuing their education, having financial independence, increased exposure to mass media along with an increased uptake of government schemes in the intervention arms as compared to comparison blocks in the midline and endline evaluations was observed as reflected in Table 1. This clearly indicated that DLTP was effective in bringing a positive change in terms of reducing school drop-outs (enhanced school retention), better livelihood, and greater utilization of government schemes (Table 4). Albeit, we cannot attribute this change completely to implementation of this DLTP only as we observed that there was a significant increase in the total household possessions at the midline and endline as compared to baseline, which might have contributed to improved digital literacy. We could not verify in the community that the increase in household possessions could be due to improved uptake of government schemes which might have resulted in a improving economic conditions of the families.

To the best of our knowledge, it is the first study to document the changes in the individual elements and overall DL following a quasi-experimental study design. The strengths of the study included that the course material of the trainings was customized with great care and sensitivity, in local languages, by experts who have dedicated experience designing content for rural populations. The trainers (or cluster coordinators) were selected from the community itself and were rigorously trained by C3 on both digital knowledge and training strategies. These trainers often went on to become key influencers of change within their communities, building sustained support around digital literacy. Also, for successfully implementing

this programme, C3 built meaningful partnerships with community health workers, who conducted regular health check-ups of the female participants as a supplementary aspect of Akanksha's trainings.

Since it was a social sector intervention study, so sample coverage and distribution got different across rounds for intervention and comparison arms which are an integral part of any community-based research. It was difficult for us to assign the intervention randomly in the chosen area of District Gumla in Jharkhand- most of the population is tribal and represent marginalized caste groups. In the initial phase of the project, it was extremely difficult to convince the families to send their school dropout girls to be a part of this DLTP. Hence, there was loss-to-follow-up in the project. Also, the girls got married off which further led to change in population across midline and endline. Hence, we tried our best to keep the background characteristics similar in the intervention and control arm across the baseline, midline and endline.

When we started this project, a digital literacy program was already in place by Google. But it was mostly for school-going children (both boys and girls). Some of the school going girls became school dropouts and got enrolled in our program by the time we rolled out the intervention. So, it was practically very difficult to assess whether the improvement in outcomes was totally attributable to our program or was a combination of multiple projects operational in that area. These can be considered limitations of the current study but controlling strict conditions in such scenarios is not possible owing to social ethics of not depriving anyone of the benefits of training program. Due to this reason, we could not apply advanced statistics to quantify the impact of DLTP. Another limitation could be that we could evaluate five elements of DL but the collaborative and productive elements could not be assessed. This was possibly due to the reason that both are more practical aspects of utilizing DL. So for evaluating these components, a study with long follow up period may be designed to see the impact of DLTP on these two elements. The way forward involves the designing of DLTPs which should focus on incorporating all the seven elements of DL with long follow-up periods.

A lot of programmatic and functional challenges along with sustainability issues were encountered while rolling out DLTP. For instance, the lack of proper digital infrastructure at both the Anganwadi and School level hindered the smooth functioning of sessions. Cluster coordinators had to often improvise while delivering the sessions with minimum tools, while students had to often share existing resources amongst themselves, making the learning process slower and more time-consuming. The girls who took part in the sessions often belonged to low-income backgrounds; their families unable to afford digital devices. Hence, they faced extreme difficulty in practically applying the knowledge they were gaining from their digital literacy classes. Among certain sections of society, there remained negative perceptions around young girls using digital technology. Either these families considered it "unsafe" for young girls to be on the internet, or they believed that social norms do not permit women to be digitally active.

To conclude, the digital literacy training programme leveraging different elements of DL led to improvement in individual elements and overall scores in the intervention arms as compared to comparison blocks. And this improvement in DL in turn led to a significant improvement in raising awareness and uptake of government schemes among adolescent girls belonging to Gumla district of Jharkhand in India. There is a felt need to scale-up such training programmes; tailor- made for every setting and weaved within cultural contexts to bridge the existing digital divide in low- and middle income countries like India.

## Author contributions

**Conceptualization:** Aparajita Gogoi, Mercy Manoranjini.

**Data curation:** Mercy Manoranjini, Mamta Gupta.

**Formal analysis:** Mamta Gupta.

**Investigation:** Aparajita Gogoi.

**Methodology:** Aparajita Gogoi, Mercy Manoranjini, Mamta Gupta.

**Project administration:** Mercy Manoranjini.

**Resources:** Aparajita Gogoi, Mercy Manoranjini.

**Supervision:** Aparajita Gogoi, Mercy Manoranjini.

**Writing – original draft:** Mamta Gupta.

**Writing – review & editing:** Aparajita Gogoi, Mercy Manoranjini, Mamta Gupta.

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
