## [Decision Letter · Decision Letter 0]

4 Nov 2024

PDIG-D-24-00341Design and implementation of digital literacy training programme: Findings of a quasi- experimental study from Rural IndiaPLOS Digital HealthDear Dr. Mercy Manoranjini, Thank you for submitting your manuscript to PLOS Digital Health. After careful consideration, we feel that it has merit but does not fully meet PLOS Digital Health's publication criteria as it currently stands. Therefore, we invite you to submit a revised version of the manuscript that addresses the points raised during the review process. Please submit your revised manuscript within 60 days Jan 03 2025 11:59PM. If you will need more time than this to complete your revisions, please reply to this message or contact the journal office at digitalhealth@plos.org. Please include the following items when submitting your revised manuscript:* A rebuttal letter that responds to each point raised by the editor and reviewer(s). You should upload this letter as a separate file labeled 'Response to Reviewers '. This file does not need to include responses to any formatting updates and technical items listed in the 'Journal Requirements' section below.* A marked-up copy of your manuscript that highlights changes made to the original version. You should upload this as a separate file labeled 'Revised Manuscript with Track Changes '.* An unmarked version of your revised paper without tracked changes. You should upload this as a separate file labeled 'Manuscript '. If you would like to make changes to your financial disclosure, competing interests statement, or data availability statement, please make these updates within the submission form at the time of resubmission. Guidelines for resubmitting your figure files are available below the reviewer comments at the end of this letter.We look forward to receiving your revised manuscript. Kind regards,Hadi GhasemiAcademic EditorPLOS Digital Health

 Hadi GhasemiAcademic EditorPLOS Digital Health

 Leo Anthony CeliEditor-in-ChiefPLOS Digital Healthorcid.org/0000-0001-6712-6626

**Journal Requirements:**

1. We ask that a manuscript source file is provided at Revision. Please upload your manuscript file as a .doc, .docx, .rtf or .tex. 

**Additional Editor Comments (if provided):**

**Reviewers' Comments:**

Reviewer's Responses to Questions

**Comments to the Author**

1. Does this manuscript meet PLOS Digital Health’s publication criteria ? Is the manuscript technically sound, and do the data support the conclusions? The manuscript must describe methodologically and ethically rigorous research with conclusions that are appropriately drawn based on the data presented.

Reviewer #1: Partly

Reviewer #2: No

2. Has the statistical analysis been performed appropriately and rigorously?

Reviewer #1: N/A

Reviewer #2: I don't know

3. Have the authors made all data underlying the findings in their manuscript fully available (please refer to the Data Availability Statement at the start of the manuscript PDF file)?

Reviewer #1: Yes

Reviewer #2: No

4. Is the manuscript presented in an intelligible fashion and written in standard English?

Reviewer #1: Yes

Reviewer #2: No

5. Review Comments to the Author

Reviewer #1: General Comments

In this manuscript, the authors  investigates the design and effects of a digital literacy training program implemented in rural India.The study utilized a quasi-experimental research design covering assessments at multiple time points (baseline, midline, and endline) to compare differences in digital literacy levels between the intervention and control arms.Overall, it is an meaningful study.However，

MINOR revision has to be done before this manuscript could be accepted for publication in the PLOS Digital Health.

Major Comments

1.The introduction section provides enough background information and also cites a lot of literature to support this study, however, the statement of the innovativeness and objectives of the study is not clear enough and I would suggest a slight modification in this regard.

2. The quasi-experimental research design used in the article may lead to differences in baseline characteristics between groups due to the lack of random assignment, so it was necessary for the authors to explain how to minimize the effects of confounding factors during the data collection process.

3.Large variations in sample size across the phases, especially at the endline assessment when the sample size of the intervention group increases while the sample size of the control group decreases, may affect the comparability of the data between the two groups, and there should be a need for more detail on sample substitution to describe the criteria for selection of the substitution sample and to demonstrate that the substitution sample is similar to the original sample in terms of its baseline characteristics, thus making the sample more representative.

4.Only univariate analyses were used in the data analysis, and multivariate analyses were not fully utilized to control for confounding factors, thus failing to reveal in greater depth the impact of the interventions on bridging the digital divide.

5.Increase the discussion of the study's limitations, especially regarding the sample's limitations and the lack of long-term impact assessment.

Minor Comments

6.The results section of the article is sometimes cited in the wrong table, e.g., on page 16 on the comparison of socio-demographic and socio-economic characteristics of the participants should be cited as Table 1 instead of Table 2. Please check that all the data descriptions match the tables.

7.On page 18, line 13, the median score for the practical aspects of the DL communication element at the midline for the intervention and control arms was 2, not 1.Please revise it.

Reviewer #2: Dear Authors,

Despite the crucial research on the relevant subject, in a first step, please revise the presentation of your work, particularly - in accordance with the publication standards - in "intelligible fashion and written in standard English."

The text leaves so much unclear, and is not appropriate for a scientific review and audience. Just some examples:

Title should be: "implementation of a digital..." / or programmes; rural India, not Rural India

Abstract: Who are "the girls" - where do they come from? What is "The project", skills about?, impact for the adolescent girls

Proportion above median?? 50%?? Unclear, what is meant actually.

p<0.000 does not exist.

Overall, please go through a proofreading revision.

6. PLOS authors have the option to publish the peer review history of their article (what does this mean? ). If published, this will include your full peer review and any attached files.

**Do you want your identity to be public for this peer review?** For information about this choice, including consent withdrawal, please see our Privacy Policy .

Reviewer #1: **Yes: ** Runsen Zhuang

Reviewer #2: No

---

## [Decision Letter · Decision Letter 1]

5 Mar 2025

Design and implementation of digital literacy training programme: Findings of a quasi- experimental study from rural India

PDIG-D-24-00341R1

Dear Mercy Manoranjini,

We are pleased to inform you that your manuscript 'Design and implementation of digital literacy training programme: Findings of a quasi- experimental study from rural India' has been provisionally accepted for publication in PLOS Digital Health.

Best regards,

Hadi Ghasemi

Academic Editor

PLOS Digital Health

**Additional Editor Comments (if provided):**

**Reviewer Comments (if any, and for reference):**

Reviewer's Responses to Questions

**Comments to the Author**

1. If the authors have adequately addressed your comments raised in a previous round of review and you feel that this manuscript is now acceptable for publication, you may indicate that here to bypass the “Comments to the Author” section, enter your conflict of interest statement in the “Confidential to Editor” section, and submit your "Accept" recommendation.

Reviewer #1: All comments have been addressed

Reviewer #3: All comments have been addressed

2. Does this manuscript meet PLOS Digital Health’s publication criteria ? Is the manuscript technically sound, and do the data support the conclusions? The manuscript must describe methodologically and ethically rigorous research with conclusions that are appropriately drawn based on the data presented.

Reviewer #1: Yes

Reviewer #3: Yes

3. Has the statistical analysis been performed appropriately and rigorously?

Reviewer #1: Yes

Reviewer #3: I don't know

4. Have the authors made all data underlying the findings in their manuscript fully available (please refer to the Data Availability Statement at the start of the manuscript PDF file)?

Reviewer #1: Yes

Reviewer #3: Yes

5. Is the manuscript presented in an intelligible fashion and written in standard English?

Reviewer #1: Yes

Reviewer #3: Yes

6. Review Comments to the Author

Reviewer #1: The revised version has addressed all my concerns.

Reviewer #3: is good

7. PLOS authors have the option to publish the peer review history of their article (what does this mean? ). If published, this will include your full peer review and any attached files.

**Do you want your identity to be public for this peer review?** For information about this choice, including consent withdrawal, please see our Privacy Policy .

Reviewer #1: No

Reviewer #3: No
